# Automatic Heart Rate Detection during Sleep Using Tracheal Audio Recordings from Wireless Acoustic Sensor

**DOI:** 10.3390/diagnostics13182914

**Published:** 2023-09-11

**Authors:** Julia Zofia Tomaszewska, Marcel Młyńczak, Apostolos Georgakis, Christos Chousidis, Magdalena Ładogórska, Wojciech Kukwa

**Affiliations:** 1School of Computing and Engineering, University of West London, London W5 5RF, UK; julia.tomaszewska@uwl.ac.uk (J.Z.T.); apostolos.georgakis@uwl.ac.uk (A.G.); 2Institute of Metrology and Biomedical Engineering, Faculty of Mechatronics, Warsaw University of Technology, 02-525 Warsaw, Poland; marcel.mlynczak@pw.edu.pl (M.M.); magdalena.ladogorska.stud@pw.edu.pl (M.Ł.); 3Department of Music and Media, Institute of Sound Recording, University of Surrey, Guildford GU2 7XH, UK; c.chousidis@surrey.ac.uk; 4Department of Otorhinolaryngology, Faculty of Medicine and Dentistry, Medical University of Warsaw, 02-091 Warsaw, Poland

**Keywords:** tracheal audio, heart rate, sleep disordered breathing, polysomnography, digital filtering

## Abstract

Background: Heart rate is an essential diagnostic parameter indicating a patient’s condition. The assessment of heart rate is also a crucial parameter in the diagnostics of various sleep disorders, including sleep apnoea, as well as sleep/wake pattern analysis. It is usually measured using an electrocardiograph (ECG)—a device monitoring the electrical activity of the heart using several electrodes attached to a patient’s upper body—or photoplethysmography (PPG). Methods: The following paper investigates an alternative method for heart rate detection and monitoring that operates on tracheal audio recordings. Datasets for this research were obtained from six participants along with ECG Holter (for validation), as well as from fifty participants undergoing a full night polysomnography testing, during which both heart rate measurements and audio recordings were acquired. Results: The presented method implements a digital filtering and peak detection algorithm applied to audio recordings obtained with a wireless sensor using a contact microphone attached in the suprasternal notch. The system was validated using ECG Holter data, achieving over 92% accuracy. Furthermore, the proposed algorithm was evaluated against whole-night polysomnography-derived HR using Bland-Altman’s plots and Pearson’s Correlation Coefficient, reaching the average of 0.82 (0.93 maximum) with 0 BPM error tolerance and 0.89 (0.97 maximum) at ±3 BPM. Conclusions: The results prove that the proposed system serves the purpose of a precise heart rate monitoring tool that can conveniently assess HR during sleep as a part of a home-based sleep disorder diagnostics process.

## 1. Introduction

The heart rate, thus monitoring the heartbeat frequency and function, is one of the most fundamental parameters in sleep medicine [1]. The analysis of this parameter becomes crucial for the diagnostics of sleep disorders in children, where breathing disorders such as sleep apnoea or snoring vastly influence the activity of the autonomic nervous system, consequently inducing rapid changes in the frequency of the heartbeat [2].

There are several methods used clinically for the detection and analysis of heart rate parameters. The practices used for heart rate monitoring during sleep include electrocardiography (ECG) and pulse oximeter plethysmography wave measurement, also known as the pulse rate. Both methods rely on the R-R interval measurement, where each “R” stands for a successive heartbeat (as part of a QRS complex). Given the recent development of wearable technology, the market offers various smart devices, most commonly in the form of a smartwatch, to provide heartbeat monitoring features ranging from heart rate calculation to atrial fibrillation detection. Nonetheless, the effectiveness and accuracy of these devices remain controversial [3].

Although ECG and the portable Holter ECG monitor are often recognized as the most efficient and successful methods for heart monitoring, the auscultation techniques are preferred for the effective diagnosis of various heart diseases; for instance, the auscultatory diagnostic of heart prosthetic valve dysfunction exhibits 92% sensitivity over other methods [4]. Such an advantage of this method can be crucial for the development of future heart rate measuring devices fully based on automated tracheal auscultation. Due to the proximity of the arteries and the heart, audio recordings collected using a contact microphone placed at the trachea could enable the isolation of recorded heart sounds, therefore presenting an opportunity to detect and calculate the heart rate. Furthermore, tracheal sound recordings can provide reliable information on breathing sounds, including snoring or apnoeic episodes, which provides an opportunity for an accurate obstructive sleep apnoea [5] diagnosis [6].

In recent years, the field of sleep medicine has seen a shift towards simpler, more personal devices for monitoring sleep and breathing during sleep. Traditionally, polysomnography (PSG) has been used to assess sleep and diagnose sleep disorders, but the use of home sleep apnoea testing (HSAT) and other personal sleep monitoring devices is becoming increasingly common [7]. The development of sensors capable of evaluating various signals simultaneously is a key aspect of this trend, as it allows for more comprehensive and accurate assessments of sleep and breathing patterns.

Many wearable devices built for assessing sleep and sleep-disordered breathing are worn on a wrist or a finger, but other approaches have also been explored. For example, sensors placed on the forehead or neck have been used to detect and analyze breathing patterns during sleep [8]. The neck is particularly promising for sensors’ placement—this location provides sensors with the ability to detect, record, and analyze each breath. Furthermore, sensors placed just above the sternum can provide important information about body position during sleep, which is a crucial parameter to analyze in sleep studies [7].

Providing that the sensor’s neck placement allows the efficient detection of breathing patterns based on audio recordings, it can also be a promising placement for heart rate evaluation [9,10]. The primary heartbeat features audible in an audio recording obtained from the neck area are the S1 and the S2 sound components [11]. The S1 and S2, also referred to as lub-dub sounds, signify the phases of the cardiac cycle associated with closing valves during systole and diastole [10]. A wireless sensor equipped with a microphone capable of detecting and recording S1 and S2 episodes could hence provide an efficient heart rate monitor alternative.

Furthermore, diagnostic tools such as PSG or ECG required for precise heart rate assessment can be frequently inaccessible in a private home environment. The potential possibility to calculate the heart rate from audio recordings obtained using a small, easily managed sensor would allow for the regular and more thorough monitoring of the patient’s health. Nonetheless, there are very few studies implementing audio recordings for the detection of heart rate (see Section 4).

Considering the reasoning presented above, the aim of this study was to introduce a heart rate calculation system based solely on heartbeat detection from tracheal audio recordings. The objective was to validate the results obtained using the proposed algorithm and wireless audio sensor against the dataset from an electrocardiogram (for initial algorithm validation), as well as whole-night polysomnography.

## 2. Materials and Methods

### 2.1. Participants

A total of 50 participants selected for this research were recruited at the Czerniakowski Hospital, Warsaw, among patients qualified for the polysomnography (PSG) study with NOX A1 system (NOX Medical Inc., Reykjavik, Iceland) (Table 1). This allowed for the simultaneous recording of patients’ heartbeats using traditional methods, as well as the desired audio output using the designed sensor. In accordance with PSG test eligibility, each participant of this study had underlying health issues related to suspected OSA.

Among others, those included loud or profuse snoring, witnessed pauses of breathing or gasping for air during sleep, morning headaches, hypertension, as well as other symptoms related to chronic nasal congestion or upper airway obstruction.

All 50 data samples collected for this study ranged from 3 to 8 h, with an average of 6 h 58 min per participant; 92% of data samples consisted of < 7 h-long recordings. Each data sample consisted of at least 3 h of audio recorded using the sensor’s contact microphone and at least 3 h of heart rate measurements obtained during the polysomnography study.

The validation study was performed on six healthy volunteers who agreed to have a simultaneous 10-min heart rate recording with an ECG Holter and acoustic sensor.

### 2.2. Sensor and Data Collection Process

For this study, the wireless acoustic sensor (Clebre, Olsztyn, Poland) was used to measure sounds from the upper respiratory tract during sleep, specifically at the level of the trachea. Two digital MEMS (micro-electro-mechanical system) microphone units were selected, both capable of 16-bit registration resolution and digital adjustments of amplification, sensitivity, and subrange. The dimensions of the entire sensor are 33 × 39 × 13 mm, and it weighs 18 g. The microphone used in the sensor is MP34DT06J (by STMicroelectronics, Amsterdam, The Netherlands). It is the MEMS audio sensor omnidirectional digital microphone, characterized by an acoustic overload point at the level of 122.5 dBSPL, 64 dB signal-to-noise ratio, –26 dBFS ±1 dB sensitivity, and very stable frequency response across the entire frequency spectrum. We decided not to implement any noise reduction mechanisms, as this might influence the signal parameters used for the analysis. The audio tract comprises a separated 2 mm audio channel; also, there is a membrane with an area of 6.25 cm^2^. The use of two microphones enabled simultaneous recording of two distinct signal sources. The first microphone captured ambient noise, while the second, in the form of a contact microphone, was utilized to record sounds directly from a body, particularly a trachea. To ensure stability throughout the night, the sensor was positioned on the subject’s neck, specifically at the sternal notch, and secured in place with medical tape (Figure 1). This configuration optimized signal quality from an analytical perspective, with the microphone, its membrane, and acoustic channel all directed toward the body during data collection.

The recorded acoustic signals were converted into a digital data stream, with a sampling frequency of 8000 Hz established as a balance between transmission restrictions and spectral content. This configuration was selected to enable accurate and reliable recordings of the upper respiratory tract sounds during sleep.

### 2.3. Signal Pre-Processing Stage

Considering the subject group comprises participants experiencing sleep breathing disorders, the majority of audio recordings obtained during data collection were highly interfered with by loud breathing sounds, such as snoring or loud body movement. To minimize the interference and distinguish all heartbeats, the frequency range of the recordings was significantly reduced by applying a bandpass filter of 20 Hz to 25 Hz, thus eliminating all frequencies above 25 Hz. Such bandpass ensures the removal of most interference while retaining the heartbeat signal that normally resides within the frequency of 20–150 Hz [12]. Although hardly audible for a human ear, the heartbeats were preserved while eliminating most of the snoring and breathing interference.

Given that heart sounds are consistently inaudible unless in direct contact with the chest area, the recording of the ambient microphone had less potential to retain any heart-related audio information. The stereo signal was split, allowing only the channel containing the recording of the contact microphone for further processing.

### 2.4. Validation

For the validation of the designed system, a dataset was developed using Holter monitor measurements. ECG Holter allows for each single heartbeat monitoring, distinguishing between heartbeat peaks and artifacts, therefore enabling the accurate calculation of true positives and false positives for each heartbeat detected by the algorithm.

A group of six healthy individuals were selected. An ECG Holter and the designed wireless sensor were placed on each participant for 10 min. For the validation dataset, the middle seven minutes of each recording were chosen to avoid any ECG Holter artifacts frequent in the first two minutes of a recording. To allow a thorough validation of the results, two independent approaches were established: the estimation of the average heart rate per minute from each device and the calculation of true positives and false positives for each heartbeat.

### 2.5. Main Algorithm

The designed algorithm begins its pre-processing by loading the chosen dataset, separating individual channels, and saving the right channel output as a mono file. To accelerate the system’s operation, the acquired mono recording is down-sampled and equalized, removing signal below 20 Hz and over 25 Hz from the frequency spectrum (as described in Section 2.3).

Considering significant variations in signal loudness, dependent on body positioning and the patient’s anatomy, the amplitude of the signal is unified by increasing the amplitude of quieter sections so that the average absolute amplitude value for each section in one recording is equal. For this purpose, the recordings are split into one-second-long sections, each one of length equal to the audio sampling rate.

To ameliorate the performance of the algorithm on all recordings within the entire dataset, the average absolute values of each signal’s amplitude are unified. All signals, unified in amplitude, are then subjected to three cycles of heartbeat peak detection. After detecting all the most prominent heartbeat peaks during the first cycle, the algorithm measures intervals between adjacent peaks and preserves only the first audible beats S1 of each heart contraction cycle, removing the second beats S2.

Subsequently, the intervals between detected S1 peaks are evaluated. The algorithm first assumes that the human heart rate does not fall below 40 beats per minute. Secondly, the distance between subsequent S1 peaks is measured. If the interspace is larger than 1.5 s and therefore corresponds to a heart rate lower than 40, the signal is re-evaluated in the second cycle with the higher sensitivity of heartbeat peak detection. The above cycle is repeated once more with slightly increasing its sensitivity.

To avoid over-calculation of heartbeat peaks and identifying peaks where they are absent, the system assumes that the highest possible human heart rate during sleep does not exceed 180 beats per minute. In its last cycle, the algorithm measures the space between adjacent peaks. If the detected heart rate exceeds 180, the heart rate is re-evaluated, readjusting the heart rate detection threshold for each minute of the recording and deleting the least prominent peaks.

### 2.6. Statistical Analysis

Before the application of the developed heart rate detection algorithm, all data recorded using the wireless acoustic sensor were analyzed to obtain the baseline descriptive statistics for all patients. For each continuous characteristic (such as weight, height, the percentage of snoring, or AHI parameter), the mean and standard deviation values were calculated; furthermore, the minimum and maximum values of each variable present in the dataset were identified (Table 1). Following the application of the proposed heart rate detection algorithm, the correlation between the heart rate detected by the developed system and the PSG-generated heart rate was evaluated. For that, Pearson’s Correlation Coefficient was implemented.

## 3. Results

The following part of the paper investigates the performance of the designed algorithm in the validation process, as well as against the polysomnography-derived heart rate, the heart rate comparison, and the true positives and false positives calculation approach for beat-to-beat detection.

### 3.1. Validation Dataset and ECG Holter

#### 3.1.1. Heart Rate

To validate the algorithm, it was investigated on a separate dataset obtained using the wireless sensor and the ECG Holter. Each 7-min-long validation recording was compared to the corresponding matrix of average heart rate measurements collected using the ECG, as shown in Figure 2. This validation approach resulted in 75% of measurements being identical or within ±1 BPM deviation, with the maximum deviation of three data points.

#### 3.1.2. True Positives and False Positives

For this testing method, a true positive indicates a heartbeat peak correctly identified by the algorithm, meaning within the same area that a peak was registered according to the ECG Holter. Such a comparison can be seen in Figure 3.

A false positive indicates a peak that was identified by the algorithm in the absence of a heartbeat, according to the ECG data. Table 2 represents all true positives and false positives detected for all validation datasets. According to the analyzed data, the average accuracy of this validation testing reaches over 92% (92.34%).

### 3.2. Study Dataset and PSG

The accuracy of the algorithm proposed was evaluated using Pearson’s correlation coefficient, achieving the average of 0.82 for all data samples, with best results of 0.93, with an error tolerance of 0 (Figure 4, Table 3). While increasing the error tolerance of ±3 beats per minute, the best results of the correlation coefficient reach nearly 0.97 (0.9664), with an average correlation coefficient of 0.89 (0.8932).

For a thorough investigation of the algorithm, the Bland-Altman analysis was used. Each recording of the tracheal audio was evaluated separately, dividing audio into 30 s windows and producing a matrix of average heart rate values for all windows (Figure 5). Obtained data was then compared to a matrix of heart rate values obtained during simultaneous polysomnography testing, with the average heart rate calculated for each 30 s of the testing time. The results of the comparison of both matrices were depicted in separate Bland-Altman plots for each subject. Figure 6 shows Bland-Altman analysis for the entire dataset tested collectively, comparing the PSG-generated HR matrix with the matrix of HR calculated using the proposed algorithm. According to Bland-Altman analysis, the coefficient of variation was equal to 4.56%, and the mean of ±2.93 SD difference between audio HR and polysomnography HR was calculated.

## 4. Discussion

Heart rate [13] is an important physiological parameter, crucial for the evaluation of sleep quality and diagnosis of various sleep disorders. HR fluctuations remain a crucial parameter in sleep medicine, as they can be used to evaluate sleep/wake patterns without the need for an electroencephalogram [13] signal [14]. Furthermore, changes in HR can be used to identify apnoeic episodes, and in patients with obstructive sleep apnoea, the detection of arrhythmias such as atrial fibrillation can be useful in monitoring and treating the condition [15].

The purpose of this study was to develop a heart rate calculation system based on heartbeat detection solely from tracheal audio recordings. The audio dataset required for this study was recorded using a wireless acoustic sensor placed at the patient’s suprasternal notch. The data for our research was obtained using a small wireless sensor with a built-in contact microphone that is to be placed just above the patient’s sternum. The Clebre sensor has already proven its ability to analyze breathing and snoring episodes, detect apnoea, and assess body positioning during sleep [16,17,18]. The portable wireless sensor used for data collection can easily be applied in a home environment for multi-night recordings, providing an opportunity for more adequate and efficient testing due to space comfortable for a patient. Adding an algorithm to detect heart rate would be highly beneficial for the sensor and the system in terms of performing sleep studies.

Subsequently, heart rate was extracted using the proposed heart rate detection and calculation algorithm and compared with heart rate recordings obtained from the simultaneous polysomnography study. The results were obtained using the correlation coefficient.

Most research on HR focuses on the photoplethysmography (PPG) signal obtained from sensors placed on a finger or, more recently, a wrist. PPG measures the changes in blood volume in the underlying tissues, which occur in response to each pulse rate. This signal can be used to derive the heart rate and other cardiovascular parameters, such as heart rate variability (HRV) and pulse wave velocity (PWV) [3]. Although wrist-worn devices are becoming increasingly popular for HR monitoring during sleep due to their non-invasive nature and convenience, their accuracy remains questionable [3].

Recent studies have shown that smartwatches and bracelets can be effective in assessing HR during sleep. In a study by Gruwez et al. [19], the accuracy of a smartwatch was compared with beat-to-beat synchronized ECG monitoring. The authors found that the correlation coefficient between smartwatch-detected HR and ECG monitoring reached over 0.9, suggesting that smartwatches could be a valuable tool for continuous photoplethysmography monitoring. In the assessment of Samsung HR detection accuracy, Sarhaddi et al. found the smartwatch had a high accuracy in detecting heart rate during both rest and exercise, with a mean absolute error of less than 5 bpm [20]. However, the accuracy of heart rate variability measurement was found to be lower, with a mean absolute error of 22.6 ms. The study suggests that smartwatches can provide accurate heart rate measurements during daily activities, but further improvements are needed for heart rate variability measurements.

A study focused on the use of home devices for HR detection investigated the clinical validation of five direct-to-consumer wearable smart devices to detect atrial fibrillation [3]. The study found that the accuracy of these devices varied significantly, and the sensitivity and specificity were dependent on the specific device. This proves the use of wearable devices as an HR detection system needs further improvement and investigation.

A novel method proposed in this research allows for heart rate detection based on the tracheal audio recordings obtained with a wearable sensor equipped with a contact microphone. A similar approach has been investigated previously by Kalkbrenner et al. [9] in a study aimed at detecting apnoea and changes in heart rate in patients with sleep-related breathing disorders. In their research, Kalkbrenner et al. used a tracheal microphone to obtain audio recordings for potential heart rate recognition and calculation. A large benefit of this algorithm was the simultaneous detection of apnoea episodes. Nonetheless, the method failed for any patients with loud breathing sounds, snoring, or those causing distribution with frequent movement. The results obtained from 10 patients showed that the tracheal sounds accurately detected apnoea and estimated heart rate during sleep with a high level of accuracy (with a correlation coefficient of 0.81) compared to PSG data.

Following a similar approach, Sharma et al. [10] introduced an algorithm for extracting the heart rate from acoustic recordings gathered at the neck level. With the application of Hilbert energy envelope and adaptive thresholding, the authors were able to identify S1 and S2 heartbeats, respectively, and consequently calculate the heart rate. The study investigates nearly 75 h of audio recordings from the neck level, from which 4482 min are then evaluated, and heart rate is extracted. The algorithm achieves a 0.93 correlation coefficient and 94.34% validation accuracy, providing a ±3.61 BPM and ±10% error tolerance. The authors identify a significant amount of sound interference (such as snoring). Thus, the algorithm applies a large amount of pre-processing, including artifact removal, low-pass filter, decimation, and median filter.

The work proposed by Kalkbrenner et al. and Sharma et al. suggests the vast potential of heart rate calculation based on tracheal audio recording. Nonetheless, both methods struggle to identify the heartbeats in a sound environment, such as snoring or change in the patient’s body positioning [9,10]. The advantage of the algorithm presented in our work is its high efficiency regardless of the extent of snoring. For instance, the correlation coefficient for a participant experiencing 68% snoring and AHI of 32.2 equals 0.92. Table 4 presents the correlation coefficients for participants with the highest snoring percentage, as well as the highest AHI. It also depicts AHI and snoring parameter values for a participant with the highest correlation coefficient for a thorough comparison.

Furthermore, the wireless sensor proposed in this work was equipped with an accelerometer, which enables the detection of body placement and activity during sleep, thereby facilitating the future exclusion of fragments disturbed by movement avoiding the recording of movement artifacts or fragments disturbed by prolonged loud snoring.

In another related study, Freycenon et al. proposed a method for estimating heart rate from tracheal sounds recorded during sleep [21]. The study involved 16 patients undergoing PSG for the diagnosis of sleep apnoea syndrome while simultaneously having their HR measured using ECG. The Authors used 1-h recording from each patient instead of the whole-night results. The study applied a deep learning approach to extract the heart rate signal from the tracheal sound recordings and found that their method achieved an accuracy between 81% and 98% in estimating heart rate, with an error rate of 5 beats per minute.

Another advantage of the system presented in this work is the highest average correlation coefficient providing 0 error tolerance, whilst the majority of related research investigates the results based on ±3 BPM error tolerance. While incrementing the error tolerance to ±3 BPM, our average correlation coefficient increased significantly, reaching 0.97 at its maximum. Furthermore, in this research, we propose the largest training dataset (over 378 h) in the current state-of-the-art. A thorough comparison of the current state-of-the-art is shown in Table 5.

In our study, 50 participants were recruited, each experiencing certain types of sleep disorders, from habitual snoring to severe obstructive sleep apnoea. From each participant, the audio data was collected using the sensor during a full-night polysomnography study.

### Study Limitations

The development of a novel wireless acoustic sensor deploying tracheal recording for accurate heart rate detection is a challenging matter due to the nature of sleeping disorders and the current golden standard for their diagnosis—PSG testing. Throughout the study, limitations were thoroughly investigated, and the methods were adjusted to address all issues.

The first recorded limitation pertained to the quality of the desired audio content. We did not study patients who would have any pathologies in their trachea or larynx. Subglottic, or tracheal stenosis, unilateral or bilateral vocal fold paralysis, could produce additional sounds that could influence the quality of our algorithms. This problem needs to be addressed in future studies. Due to the character of sleeping disorders, the majority of research subjects were experiencing various breathing difficulties, such as snoring, which strongly interferes with the desired audio signal. To ensure the maximum preservation of a heartbeat sound, the according frequency bandpass was implemented, eliminating the vast majority of interference. However, provided that snoring causes a vibrational response of the vocal tract, parts of the filtered signal remained affected by the snoring activity. Nonetheless, the sound of a heartbeat overlapping disturbances caused by snoring was recorded, causing subtle regular changes in the signal’s amplitude.

Considering that the placement of the sensor changes slightly depending on the patient’s body positioning during sleep, certain fragments can be significantly decreased in loudness. Furthermore, the volume of audio recordings varies significantly depending on each subject’s anatomical structure; in the case of a patient’s obesity, the presence of a larger amount of tissue can lead to a drop in a heartbeat’s loudness. To reduce this factor’s impact on the algorithm’s workflow, the amplitude of each signal is averaged and normalized equally for each subject. Future studies may increase our knowledge about the differences in the loudness of breathing vs. heart rate signals in accordance with the sleep position. This is because our sensor detects the sleep position and can allow such analysis. Also, thanks to the accelerometer used, we can remove signal analysis during body movement when the acoustic noise is intensive.

To ensure the correct detection of all present heartbeats while avoiding the incorrect interpretation of signal peaks where a heartbeat is absent, it was crucial to determine the maximum and minimum values of beats per minute. According to medical literature, the maximum human heart rate cannot exceed 220 beats per minute, regardless of age [22], whereas a heart rate falling below 60 beats per minute is recognized as bradycardia [23]. However, considering this research pertains to asleep individuals, a slight decline in heart rate is expected. For the purposes of this research, the specific minimal and maximal heart rate of 40 and 180 beats per minute was established, respectively. This is because we did not enroll patients with any known cardiac arrhythmias. Additional studies on patients with irregular heart rhythms are needed to show the performance of this algorithm in such cases.

## 5. Conclusions

Audio recordings of heart rate seem much easier and more user-friendly in the home environment than the ECG. This study of heart rate detection and calculation based on tracheal audio recordings proves the high accuracy and reliability of the proposed method. Employing the proposed wireless acoustic sensor and the developed heart rate calculation algorithm provides a promising result and a solid base for further investigation. Considering the dataset for this study was obtained by placing the wireless acoustic sensor at the suprasternal notch, the next step in this research includes the comparison of the algorithm-generated results depending on the placement of the sensor—suprasternal notch versus its placement at the heart level. Subsequent future research efforts will include a real-time application of the algorithm, as well as a possible thorough investigation of sleep/wake patterns based on combined algorithms analyzing heart rate, breathing parameters, and body position.

## Figures and Tables

**Figure 1 diagnostics-13-02914-f001:**
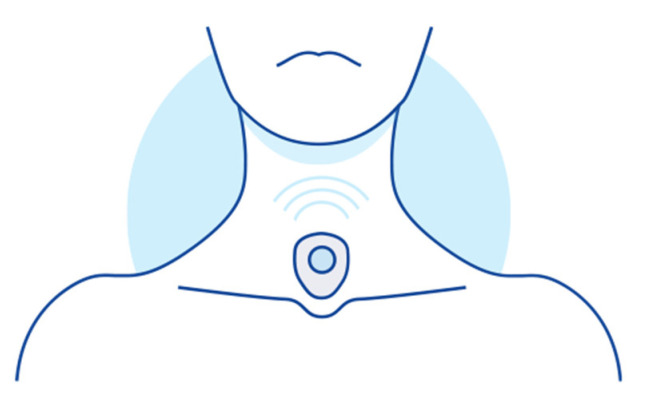
The placement of the wireless acoustic sensor.

**Figure 2 diagnostics-13-02914-f002:**
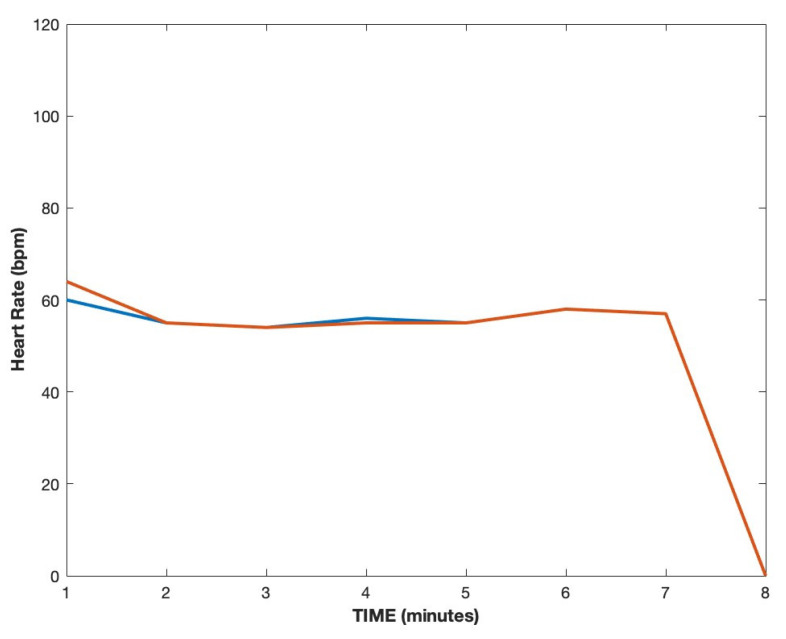
Comparison of ECG heart rate measurements (blue) with the heart rate acquired using the wireless sensor and the proposed algorithm (orange)—data shown for a single participant.

**Figure 3 diagnostics-13-02914-f003:**
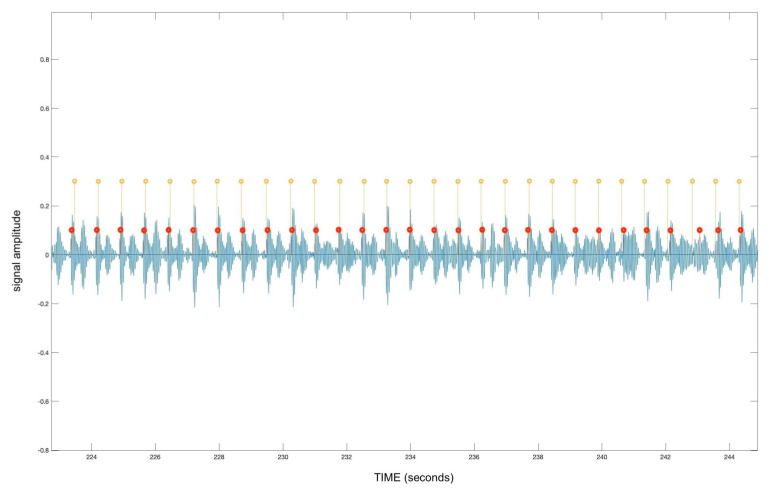
Comparison of audio recording (blue), peaks detected by the algorithm (red), and the peaks indicated by the ECG dataset (yellow)—data shown for a single participant.

**Figure 4 diagnostics-13-02914-f004:**
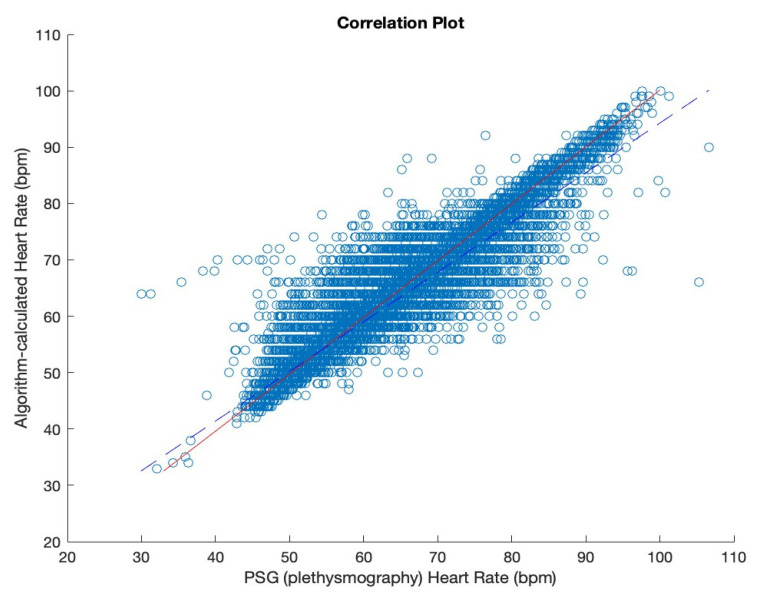
Correlation plot depicting the relation between the heart rate obtained from polysomnography (*x*-axis) and the heart rate calculated by the algorithm designed in this study (*y*-axis).

**Figure 5 diagnostics-13-02914-f005:**
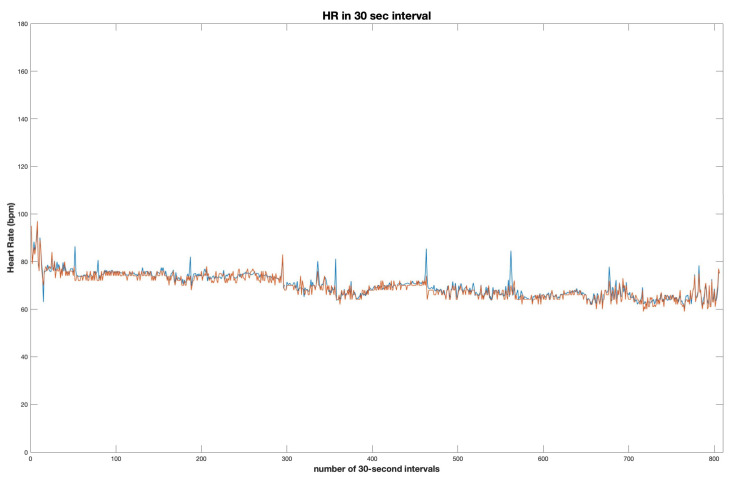
An example of PSG-derived heart rate (blue) compared to the algorithm-generated heart rate (orange)—data shown for a single participant.

**Figure 6 diagnostics-13-02914-f006:**
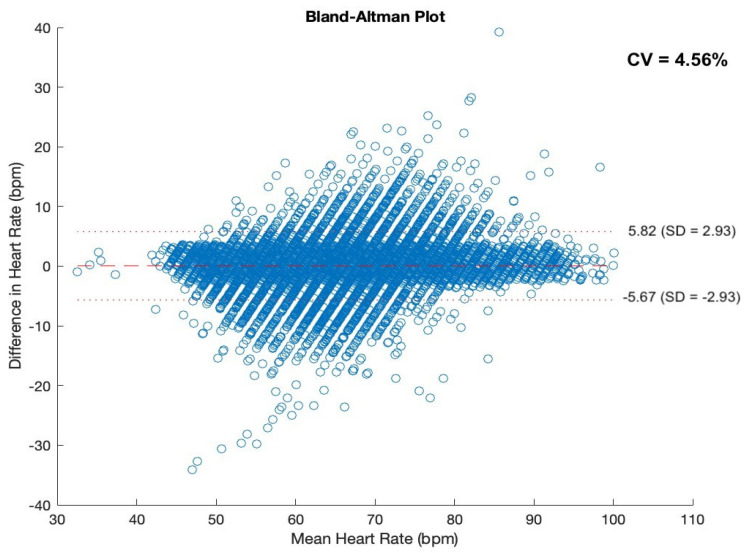
Bland-Altman analysis plot depicting the relation between the heart rate obtained from polysomnography and the heart rate calculated by the algorithm designed in this study.

**Table 1 diagnostics-13-02914-t001:** Demographic overview of the research group involved in the study, depicting minimum and maximum values within the dataset, as well as mean and standard deviation values for the entire dataset. TRT—Total Recording Time. TST—Total Sleep Time. AHI—Apnoea-Hypopnoea Index. ODI—Oxygen Desaturation Index.

Parameter	Males	Females	Total	Mean-Max for Total
Participants Number	33	17	50	50
Age	49.15 ± 13.1	60.12 ± 11.5	52.88 ± 13.5	29.0–77.0
Height (cm)	177.09 ± 5.9	161.53 ± 8.3	171.80 ± 10.0	150.0–186.0
Weight (kg)	100.00 ± 22.2	80.71 ± 21.2	93.44 ± 23.5	53.0–158.0
TRT	07:00:11 ± 0.0	06:55:39 ± 0.1	06:58:38 ± 0.0	03:05:00–08:14:00
TST	06:06:40 ± 0.0	05:33:18 ± 0.1	05:55:19 ± 0.0	02:25:00–07:35:00
AHI	29.66 ± 19.6	26.86 ± 22.2	28.71 ± 20.3	2.4–93.3
ODI	27.79 ± 19.5	25.15 ± 23.3	26.90 ± 20.7	1.4–93.9
Snoring (%)	40.79 ± 20.6	35.92 ± 19.8	39.14 ± 20.2	0.8–80.1
Time % <90% SaO_2_	13.57 ± 17.4	16.48 ± 24.7	14.56 ± 20.0	0.0–67.9

**Table 2 diagnostics-13-02914-t002:** True positives and false positives are calculated for each validation dataset.

Validation Dataset Nr	Total Number of S1 Peaks Detected	True Positives (TP)	False Positives (FP)	TP %
1	484	419	65	86.57
2	559	508	51	90.88
3	450	401	49	89.11
4	744	711	33	95.56
5	517	501	16	96.91
6	422	401	21	95.02
MEAN	529.33	490.17	39.17	92.34

**Table 3 diagnostics-13-02914-t003:** Correlation Coefficient for Spearman’s and Pearson’s Correlation at ±0 BPM error tolerance. The last column represents the correlation coefficient at ±3 BPM.

Spearman’s Correlation Coefficient	Pearson’s Correlation Coefficient	Correlation Coefficient at ±3 BPM Error Tolerance
0.8025 ± 0.088	0.8203 ± 0.076	0.8932 ± 0.069

**Table 4 diagnostics-13-02914-t004:** AHI, snoring parameter, and correlation coefficient were calculated for participants with the highest AHI within the dataset, highest snoring within the dataset, and highest correlation coefficient, respectively.

Participant with:	Gender	Age	AHI	Snoring %	Correlation Coefficient
Highest AHI	Male	34	93.3	46.8%	0.90
Highest Snoring %	Female	76	54.6	80.1%	0.76
Highest Correlation Coefficient	Female	44	14.2	34.0%	0.93

**Table 5 diagnostics-13-02914-t005:** Summary of State-Of-The-Art methods for estimation of heart rate from tracheal audio recordings.

Literature	Proposed Method	Validation Method	Number of Subjects	Recording Time	Results	Advantages of the Method
Kalkbrenner et al., 2017 [9]	Signal amplification	Subjects undergoing full night polysomnography correlated with developed system	10	69 h 45 min	0.816 correlation coefficient	High correlation coefficient (relative to currently presented work).
	10–50 Hz bandpass filter					
	Peak detection					
Frecenon et al., 2021 [21]	Adaptive Prediction Filter	ECG signal as a reference method for validation subject.	Study: 16	Study: 16 h	Study: average at ±3 BPM = 81–98%	High validation results (relative to currently presented work).
	Cross-correlation		Validation: 1	Validation: 8 h 30 min	Valid.: average at ±3 BPM = 94.5%	
Sharma et al., 2018 [10]	Hilbert energy envelope	Subjects undergoing polysomnography correlated with developed system	13	75 h	Accuracy at ±3.61 BPM = 94.34%	High validation results provide a relatively large dataset.
	Adaptive thresholding				Correlation coefficient at ±3.61 BPM = 0.93	
THIS WORK	Peak Normalisation	Heart rate calculated by proposed algorithm compared with simultaneous ECG measurements—False Positive and True Positive calculation.	Study: 50	Study: 378 h 40 min	Study: average correlation coefficient at zero error tolerance (±0 BPM) = 0.82, highest correlation coefficient at zero error tolerance = 0.93	Highest correlation coefficient, providing 0 error tolerance (±0 BPM).
	20–25 Hz bandpass filter					Largest study dataset in the state-of-the-art.
	Peak detection		Validation: 6	Validation: 42 min	Validation: 92.34%	Largest validation dataset in the state-of-the-art.
	S1 peak preservation, S2 peak rejection					S1 and S2 peak detection and differentiation.

## Data Availability

Data supporting the reported results are available from the corresponding author on reasonable request.

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
