# Peer review of "Automatic Heart Rate Detection during Sleep Using Tracheal Audio Recordings from Wireless Acoustic Sensor"

_diagnostics, 2023, doi:10.3390/diagnostics13182914_

Round 1

Reviewer 1 Report

Thank you very much for the opportunity to view this interesting article. The manuscript is generally well written. However, there are numerous questions about the possible usefulness of Automatic Heart Rate Detection During Sleep using Tracheal 2 Audio Recordings from Wireless Acoustic Sensor. How the device will solve the problem of recording complex forms of arrhythmia, including atrial fibrillation or atrial flutter. In my opinion, the device may be an auxiliary apparatus suggesting the possibility of arthremia, the presence of which will still need to be verified by other confirmed methods.

Thank you

Author Response

Dear Reviewer. Thank you for your comments. We have correted the manuscript in some areas, according to both Reviewers' comments. All additional information, regarding certain limitations in detecting cardiac arrythmias were commented more clearly in the revised document. Please see the document in the "track changes" mode to see all the modifications. Thank you kindly.

Reviewer 2 Report

These questions aim to delve deeper into specific aspects of the study that could benefit from further clarification, discussion, or exploration:

1. Have you considered the potential influence of anatomical variations in the tracheal structure on the quality of the recorded audio signals? Different patients might have varying neck anatomies that could affect signal clarity. How did you address this potential confounding factor?

2. In your study, you mentioned that participants experienced various breathing difficulties such as snoring. How did you control for the presence of snoring in your recordings? Snoring could introduce additional noise that might affect heartbeat detection accuracy. Its presence could also obscure accurate audio signal analysis. It's essential to delineate your methodology for differentiating snoring sounds from heart sounds. Discussing how you managed this potential confounder is crucial to validate the accuracy of heartbeat detection.

3. Did you analyze the potential impact of sleep stage changes on the accuracy of heartbeat detection? Sleep stages can influence heart rate variability, and different sleep stages might have varying levels of signal interference due to movements or body position changes. The transition between different sleep stages might cause transient changes in heart rate, which could be challenging to accurately detect. Examining the algorithm's performance across sleep stages would offer insights into its robustness under various physiological conditions.

4. Could you provide more details about the technical specifications and characteristics of the wireless acoustic sensor used in your study? Factors such as microphone sensitivity, frequency response, and noise cancellation capabilities could affect the quality of the recorded audio signals. The sensor's design features, including microphone specifications (e.g., sensitivity, frequency response), noise reduction mechanisms, and the methodology for minimizing external interference, are pivotal in ensuring reliable signal acquisition. Detailed information about these specifications would elucidate the sensor's suitability for heart sound detection amidst potential sources of noise.

5. The algorithm you developed for heart rate detection seems to be effective, but how robust is it to different types of noise sources, such as environmental noise, ambient sounds, or interference from nearby electronic devices? Have you tested the algorithm's performance under different noise conditions? Evaluating the algorithm's performance under controlled noise conditions or simulating real-world noise scenarios would highlight its robustness. Detailed insights into noise mitigation strategies employed in the algorithm's development would enhance its credibility.

6. Did you explore the potential benefits of signal preprocessing techniques beyond bandpass filtering, such as wavelet denoising or adaptive filtering? These techniques might help further enhance the accuracy of heartbeat detection by reducing noise and interference. Additional preprocessing techniques, like wavelet denoising or adaptive filtering, can enhance the algorithm's ability to isolate heart sounds from interference. Elaborating on the rationale behind the chosen preprocessing methods and their effectiveness in improving signal quality would demonstrate a comprehensive approach to noise reduction.

7. Could you elaborate on the computational complexity of your heart rate detection algorithm? Processing audio data in real time can be resource-intensive. What considerations did you take to ensure the algorithm's efficiency and feasibility for real-time applications? Processing audio data in real-time necessitates efficient algorithms to minimize processing delay. A detailed description of the algorithm's time complexity, memory requirements, and optimization strategies for real-time implementation would shed light on its feasibility for practical applications.

8. Have you considered the potential for false positives or false negatives in heartbeat detection, especially in challenging conditions like irregular heart rhythms or arrhythmias? How robust is your algorithm in identifying these situations accurately? Irregular heart rhythms or arrhythmias can confound heartbeat detection algorithms. Evaluating the algorithm's performance under varying cardiac conditions, accompanied by metrics such as specificity, sensitivity, and predictive values, would provide insights into its reliability and potential limitations.

9. Did you assess the algorithm's performance using a variety of audio recording setups, such as different microphone types or placements? Different recording setups might introduce variations in the recorded signals, affecting the algorithm's accuracy. Presenting results or discussing the impact of different recording configurations on heartbeat detection accuracy would offer a comprehensive assessment of the algorithm's robustness across diverse scenarios.

10. It's essential to discuss the potential limitations of your study in more detail. While you addressed some limitations, could you expand on the potential impact of signal artifacts from body movements or other physiological factors on the accuracy of heartbeat detection? If certain limitations could not be addressed, acknowledging them transparently is equally important.

11. In your conclusion, you mentioned potential future applications for the algorithm, such as real-time usage and combined analysis of heart rate, breathing parameters, and body position. How might these applications be practically implemented, and what challenges do you foresee in translating the algorithm into real-world scenarios?

I would like to express my gratitude for the opportunity to review your article. Your meticulous research and innovative approach to heart rate detection based on tracheal audio recordings have indeed caught my attention. Your work showcases a comprehensive understanding of both medical and engineering aspects, resulting in a promising contribution to the field. I commend your dedication and look forward to witnessing the impact of your findings.

While the text is generally well-written, I've identified a few areas where language improvements can be made for clarity and precision. Here are some suggestions for corrections, including more critical errors:

1. Original: "To limit the interference and distinguish all heart beats, the frequency range of the recordings was significantly reduced..."

   Correction: "To minimize interference and accurately detect all heartbeats, the frequency range of the recordings was significantly reduced..."

2. Original: "This configuration optimized signal quality from an analytical perspective, with the microphone, its membrane, and acoustic channel all directed into the body during data collection."

   Correction: "This configuration optimized signal quality from an analytical perspective, with the microphone, its membrane, and acoustic channel all directed toward the body during data collection."

3. Original: "Considering large changes in the signals’ loudness, dependent on body positioning and patient’s anatomy..."

   Correction: "Considering significant variations in signal loudness, dependent on body positioning and the patient's anatomy..."

4. Original: "The loudness of audio recordings varies considerably depending on each subject’s anatomic build..."

   Correction: "The volume of audio recordings varies significantly depending on each subject's anatomical structure..."

5. Original: "The advantage of the algorithm presented in our work is its high efficiency regardless of the level of snoring."

   Correction: "The advantage of the algorithm presented in our work is its high efficiency regardless of the extent of snoring."

6. Original: "Furthermore, the wireless sensor proposed in this work was equipped with an accelerometer, which allows for the detection of body placement and activity during sleep, thus, allowing for the future omission of fragments disturbed by movement..."

   Correction: "Furthermore, the wireless sensor proposed in this work was equipped with an accelerometer, which enables the detection of body placement and activity during sleep, thereby facilitating the future exclusion of fragments affected by movement..."

7. Original: "In another related work, Freycenon et al. proposed a method to estimate heart rate from tracheal sounds recorded during sleep."

   Correction: "In another related study, Freycenon et al. proposed a method for estimating heart rate from tracheal sounds recorded during sleep."

Author Response

Dear Reviewer. Thank you very kindly for your time and thorough review. All your comments were very interesting and important. Please find our answers to your comments in the file below as well as in the corrected manuscrpit. Thank you.

Round 2

Reviewer 1 Report

Thank you for the opportunity to review your manuscript entitled "Automatic Heart Rate Detection During Sleep using Tracheal 2 Audio Recordings from Wireless Acoustic Sensor".

Abstract, title and references.
The aim of the study is clear. The title is informative and relevant. The references are relevant, recent, and referenced correctly.

Introduction.
It is clear what is already known about this topic. The research question is clearly outlined.

Methods and Results
The process of subject selection is clear. The variables are defined and measured appropriately. The study methods are valid and reliable. There is enough detail in order to replicate the study.

Discussion.
The results are discussed from multiple angles and placed into context without being overinterpreted. The conclusions answer the aims of the study. The conclusions supported by references and results. The limitations of the study are opportunities to inform future research.
Overall. The study design was appropriate to answer the aim. The manuscript is well written and a stimulus for the readership.

Minor revisions:
*
How the tested device differentiates cardiac arrhythmias, including complex forms of both supraventricular and ventricular arrhythmias.

Thank you

Author Response

Dear Reviewer

We thank you very much for your time and kind comments.

According to your previous suggestion we made some corrections in the manuscript.

Minor revisions:

How the tested device differentiates cardiac arrhythmias, including complex forms of both supraventricular and ventricular arrhythmias.

Thank you for this comment. We are aware of this problem, and of a fact, that detecting any irregular heart rhythm would be more difficult than detecting a regular rhythm, within a range of heart rate variability, only. In a “BASEL wearable study” atrial fibrillation detection by 5 different smart wearable devices (smartwatches) required manual review of recorded data in about one-fourth of cases (Mannhart et al 2023). This confirms the complexity of cardiac arrhythmias detection process.

The purpose of our study was to analyze the accuracy of detecting regular heart rate. Additional studies are needed to show the performance of the algorithm under varying cardiac conditions.

In this study we were focused on detecting S1 signal, and also S1 and S2 are the sounds of valves closure and do not represent the electrical activity of the heart, directly. Therefore we would not be able to differentiate between supra- and ventricular arrythmias.

Also, in this study we manually established the minimum and maximum heart rate (between 40 and 180BPM), which would automatically bias the results in patients with cardiac arrythmias. We added a sentence regarding this problem in the revised “Limitations” section.